# Slotted Monopole Patch Antenna for Microwave-Based Head Imaging Applications

**DOI:** 10.3390/s22197235

**Published:** 2022-09-23

**Authors:** Abdulrahman Alqahtani, Mohammad Tariqul Islam, Md Siam Talukder, Md Samsuzzaman, Mohsen Bakouri, Sofiene Mansouri, Thamer Almoneef, Socrates Dokos, Yousef Alharbi

**Affiliations:** 1Department of Medical Equipment Technology, College of Applied, Medical Science, Majmaah University, Majmaah City 11952, Saudi Arabia; 2Department of Biomedical Technology, College of Applied Medical Sciences in Al-Kharj, Prince Sattam Bin Abdulaziz University, Al-Kharj 11942, Saudi Arabia; 3Centre for Advanced Electronic and Communication Engineering, Department of Electrical, Electronic and Systems Engineering, Faculty of Engineering & Built Environment, Universiti Kebangsaan Malaysia (UKM), Bangi 43600, Malaysia; 4Department of Computer and Communication Engineering, Faculty of Computer Science and Engineering, Patuakhali Science and Technology, Patuakhali 8602, Bangladesh; 5Department of Physics, College of Arts, Fezzan University, Traghen City 71340, Libya; 6Laboratory of Biophysics and Medical Technologies, Higher Institute of Medical Technologies of Tunis, University of Tunis El Manar, Tunis 1068, Tunisia; 7Electrical Engineering Department, College of Engineering, Prince Sattam Bin Abdulaziz University, Al-Kharj 16278, Saudi Arabia; 8Graduate School of Biomedical Engineering, University of New South Wales, Sydney, NSW 2052, Australia

**Keywords:** monopole patch antenna, microwave-based, stroke recognition, microwave imaging

## Abstract

A modified monopole patch antenna for microwave-based hemorrhagic or ischemic stroke recognition is presented in this article. The designed antenna is fabricated on a cost-effective FR-4 lossy material with a 0.02 loss tangent and 4.4 dielectric constant. Its overall dimensions are 0.32 λ × 0.28 λ × 0.007 λ, where λ is the lower bandwidth 1.3 GHz frequency wavelength. An inset feeding approach is utilized to feed the antenna to reduce the input impedance (z = voltage/current). A total bandwidth (below −10 dB) of 2.4 GHz (1.3–3.7 GHz) is achieved with an effective peak gain of over 6 dBi and an efficiency of over 90%. A time-domain analysis confirms that the antenna produces minimal signal distortion. Simulated and experimental findings share a lot of similarities. Brain tissue is penetrated by the antenna to a satisfactory degree, while still exhibiting a safe specific absorption rate (SAR). The maximum SAR value measured for the head model is constrained to be equal to or below 0.1409 W/kg over the entire usable frequency band. Evaluation of theoretical and experimental evidence indicates the intended antenna is appropriate for Microwave Imaging (MWI) applications.

## 1. Introduction

Stroke is a leading cause of permanent disability and death worldwide, accounting for more than five million deaths, with another five million permanently disabled every year [1,2]. The total European cost was estimated to be 64.1 billion EUR in 2010 [1]. Stroke is commonly classified into two types: (1) hemorrhagic, accounting for 85% of all stroke cases and (2) ischemic, accounting for 15%, each with its own set of treatment options [1,3]. Mismatched treatment has disastrous consequences, necessitating the detection of the stroke type before treatment. The first 4.5 h after an ischemic stroke are considered “the golden period” for receiving thrombolytic therapy, according to the American Stroke Association and clinical thrombolysis guidelines [4]. However, if thrombolytic therapy is administered to patients who have a hemorrhagic stroke, the outcome is disastrous. Currently, only 10% of patients with ischemic stroke have received thrombolytic therapy during the “golden period” [5]. As a result, a portable, quick, and simple-to-use device is required for pre-hospital stroke type diagnosis and acute-phase stroke monitoring.

Magnetic Resonance Imaging (MRI), Computed Tomography (CT), and Positron Emission Tomography (PET) are the most frequently used techniques for diagnosing stroke [2,6,7,8]. While these imaging methods can produce highly precise images of the human brain, they do have certain limitations. MRI is particularly contraindicated in patients who have metal biomedical implants. CT is ionizing radiation. PET imaging is performed by injecting radioactive materials into the patient’s body [8]. Additionally, each of these methods is costly, time-consuming, and does not utilize portable devices [2,7]. According to the World Health Organization (WHO), about three-quarters of the world population does not have access to reliable and affordable medical imaging systems [6]. Hence, there is a need for an alternative imaging technology that can provide reliable identification of brain abnormalities, while being safe, rapid, economical, and portable.

Recently, Microwave Imaging (MWI) has emerged as a viable technique for brain imaging [1,4,5,6]. MWI provides a safe, affordable, non-ionizing, portable, non-invasive, and quick imaging method for the identification of brain illness. In comparison to conventional imaging technologies, this novel technology produces high-quality images of the brain and can be utilized for continuous brain monitoring. MWI is a viable option for brain imaging in poor countries and rural locations because of its cost-effectiveness and reasonable image quality. Numerous MWI devices and prototypes have been proposed recently. Among these, the Strokefinder, created by Medfield Diagnostics, and the EMTensor BrainScanner are two prominent examples (both of which are now being tested on humans) [2,4,9].

Generally, the antenna array is one of the main hardware components of an electromagnetic imaging system, and thus the system’s success is heavily reliant on the performance of the antenna arrays [10]. However, quantifying the requirements of transmitting and receiving antennas for biomedical applications of electromagnetic imaging is dependent on the type of imaging algorithm, as well as the receiving system’s target dynamic range and sensitivity [10]. The radar-based algorithm, for example, detects and locates a target within the imaged domain based on the time delay between transmitted and received signals [11]. As a result, this method necessitates the use of wideband antennas to obtain a high-resolution image [12]. Various designs of broadband slot antennas have been proposed in recent years to address the narrow impedance and frequency operating bandwidth issues [13,14,15]. Depending on the application, the frequency band used with this technique ranges from 0.3 to 11 GHz [10]. For example, the method uses 1 to 11 GHz frequencies to image the breast [11,12], whereas 1 to 4 GHz frequency bands are used to reconstruct the image of the human head [16,17]. Additionally, multiband operating frequencies could reach 16 GHz for an internet of things in smart healthcare applications [18]. On the other hand, tomography-based algorithms typically use a single frequency to reconstruct the dielectric properties of the imaged domain [19,20]; however, recent tomography algorithms use multiple frequencies [21]. In this case, the imaging antenna for multiple-frequency tomography should operate at multiple or wideband frequencies, allowing the optimal frequencies to be selected during the image reconstruction process.

Several designs of ultra-wideband (UWB) antennas have been proposed to address the optimal settings for head microwave imaging systems. A compact 3D UWB antenna has been fabricated with operational frequencies from 1.1 to 2.2 GHz, with a 67% fractional bandwidth [22]. A recent study proposed a wide-frequency-range antenna with a 1.82 GHz operational bandwidth and peak gain of 5 dBi [23]. Another study proposed an UWB antenna with a frequency bandwidth of 2.37 GHz, peak gain of 5.95 GHz, and fidelity factor over 90% [24].

This study modeled and fabricated an UWB antenna suitable for brain stroke microwave imaging. This microstrip antenna, including a slotted ground, exhibited promising simulation and measurement results. The proposed antenna (i) is fabricated using cost-effective FR-4 lossy material with a 0.02 loss tangent and 4.4 dielectric constant. (ii) Its overall dimensions are 0.32 λ × 0.28 λ × 0.007 λ, where λ is the lower bandwidth 1.3 GHz frequency wavelength. (iii) The acquired bandwidth spans 1.3 to 3.7 GHz, which is relatively wider than the reported antennas in the literature [10,22,24,25,26,27,28,29,30,31]. (iv) In comparison with reported ranges in previously published work [22,24,27,30,31], the proposed antenna offeres an effective peak gain of over 6.15 dBi and an efficiency of over 90%. Additionally, (v) the maximum measured SAR value for the head model is constrained to be equal to or below 0.1409 W/kg over the entire usable frequency band, which is less than the values reported in the literature [24,25,28,31]. Along with the ease of its antenna design and use of affordable materials, it can readily be utilized for head microwave imaging.

## 2. Materials and Methods

### 2.1. Initial Antenna Design

The antenna is constructed on an FR-4 epoxy dielectric substrate with a thickness of 1.5 mm, dielectric constant of 4.3, and a loss tangent of 0.02. A 50 Ω microstrip line supplies power to the antenna. The following formulas were used to determine the initial patch width *W* and patch length *L*, which were 30 and 25 mm, respectively. The resonance frequency was set to 2.5 GHz. Equations used to determine the initial dimensions of the antenna radiator’s width *W* and length *L* were:(1)$$W=\frac{c}{2f_{r}}\sqrt{\frac{2}{\varepsilon_{r}+1}}$$
(2)$$\varepsilon_{reff}=\frac{\varepsilon_{r}+1}{2}+\frac{\varepsilon_{r}-1}{2}{\left[1+12\frac{h}{W}\right]}^{-\frac{1}{2}}$$
(3)$$L=\frac{c}{2f_{r}\sqrt{\varepsilon_{reff}}}$$
where W refers to the patch width, c refers to vacuum light speed, $f_{r}$ is the resonant frequency, $\varepsilon_{r}$ is the relative dielectric permittivity, $\varepsilon_{reff}$ is the effective permittivity of the substrate, h is the substrate height, and L is the patch length.

It is important to consider the transmission line width ($W_{f}$) when calculating radiation intensity because it directly impacts the signal’s bandwidth. By plugging in values for the desired impedance ($z_{o}$), thickness of copper (t), dielectric thickness (h), and relative dielectric constant (∈r), the fundamental Equation (4) can be used to calculate the width microstrip line.
(4)$$W_{f}=\frac{7.48\times h}{e^{\left(\frac{z_{o}\times \sqrt{\in r\times 1.41}}{87}\right)}}-1.25\times t$$

In accordance with the fundamental equation, the width of the transmission line was calculated to be 2.7 mm when the impedance is 50 ohms, copper thickness is 0.035 mm, dielectric thickness is 1.5 mm, and dielectric constant is 4.4.

By using the basic monopole Equations (1)–(3), the antenna’s initial dimensions were derived at the beginning of the design. Afterward, we analyzed the various parameters through a parametric sweep operation and developed the proposed ground and patch shapes using the try-and-error method.

### 2.2. Parametric Study

Figure 1 illustrates the process of various modifications of the initial design and the final configuration of the proposed antenna (a–e). The impact of the patch and ground plane on antenna performance was evaluated step by step by fitting various types of modifications. Referring to Figure 1a, the main purpose that a good impedance matching was not achieved when using an edge-fed patch is that, at the edge of the patch, the input impedance is approximately 200 ohms, while the source impedance is 50 ohms. One can use a quarter wavelength transformer (QWT) to transform the 200 ohms to 50 ohms; however, the use of a QWT limits the bandwidth of the antenna response and increases the size of the antenna. The device with the same full patch and partial ground (Figure 1b) revealed a fundamental resonance mode at around 2.6 GHz, reaching more than –14 dB with a range of operation of 1.1 GHz (2.2 to 3.3 GHz). Since the antenna with slotted ground produced an active operating band, we chose the slotted ground plane for further modification, as shown in Figure 1c. As a result, this configuration extended the operational area band from 1.2 to 3 GHz, with a resonance frequency of 1.8 GHz. The notch in the slotted ground parallel to the feed line of the patch was the fourth modification, as displayed in Figure 1d. This geometry increased the antenna bandwidth to 1.2–3.1 GHz with multiple resonance frequencies. The final modification stage was changing the patch dimensions, as illustrated in Figure 1e, resulting in enlarging the antenna bandwidth to 1.3–3.7 GHz, as shown in Figure 2. The combination of using inset-fed patch along with the slot in the ground allowed for better matching, wider bandwidth, and a relatively smaller size than if we had to use a QWT. The human head is composed of complex materials with various electrical properties (e.g., primitivity and conductivity). Such constitutive parameters are dispersive, with frequency-dependent responses. Therefore, in this paper, an antenna with multiple resonance frequencies is needed. Such a multi-resonance response allows for enhanced imaging results when processing the collected data.

### 2.3. The Proposed Antenna Design

Various modifications (Figure 1) were made to achieve the best antenna design suitable for medical imaging. The final patch dimensions of the proposed design were 20 mm for the width (W) and 25 mm for the length (L). The proposed antenna’s geometric layout is depicted in Figure 3, featuring a simpler inset-fed patch and a slotted ground plane with a modest parallel notch for the feed line. The antenna’s overall dimensions are roughly $70\times 60\times 1.5\;{\mathrm{mm}}^{3}$. The feedline has a width of 2.72 mm and a height of 19.5 mm. Fine-tuned design settings were made using COMSOL Multiphysics finite element software (COMSOL 5.6 AB, Stockholm, Sweden) to achieve optimal bandwidth, gain, and performance. Table 1 contains all the design-optimized parameters of the proposed antenna.

## 3. Results and Discussion

### 3.1. Surface Current Analysis

Antenna surface currents are shown in Figure 4a,b at two significant frequencies, 1.8 GHz and 3.3 GHz, respectively. Based on Figure 4a, the antenna’s strongest conducting regions are the feedline and its surroundings, the lateral edges of the ground plane, the square-shaped patch, and a certain amount of current distributed over the top and bottom sides of the slotted ground plane. When the frequency was increased, current conduction regions shifted significantly. According to Figure 4a, low resonance at 1.8 GHz is greatly affected by the surface current at the upper feedline portion and the bottom lateral edges of the ground plane. Due to rotational surface currents in the radiation element, the rectangular slot on the ground plane here produces additional polarization. An improved performance was therefore achieved, including improved bandwidth, improved radiation pattern polarization, stable radiation pattern properties, and resonance at a lower frequency. The circulating current along the bottom edge of the patch radiator, as well as the feedline gap and the top corner of the slotted ground, is responsible for the upper resonance mode at 3.3 GHz (Figure 4b). The higher-order flow of current left lower surface currents on the radiating patch and the ground. Despite this, the antenna maintains its balanced periodic movements over the patch and ground to accomplish frequency widening.

### 3.2. Antenna Fabrication and Frequency Domain Measurements

To evaluate and assess the outputs of the antenna structure, three of the most popular simulation platforms were used: HFSS (ANSYS, Canonsburg, PA, USA), CST Microwave Studio (Dassault Systemes, Vélizy-Villacoublay, France), and COMSOL Multiphysics. Fabrication of the design structure was performed, and experimental measurements of the S-parameter (−10 dB impedance bandwidth) were obtained. Figure 5a illustrates the front and back views of the prototype and Figure 5b shows the impedance measurement setup. The anechoic chamber shown in Figure 5c was used to conduct in-depth measurements of the antenna gain, radiation efficiency, and radiation pattern. The value of the S-parameters of the prototype was measured using a PNA network analyzer (Keysight Technologies, Santa Rosa, CA, USA: model N5227A 10 MHz–67 GHz), with measurements included in Figure 5c. Graphs of the simulated and measured data were also plotted using OriginPro (OriginLab, Northampton, MA, USA). With a maximum impedance of −10 dB, the measurement of the proposed antenna has a bandwidth of 2.45 GHz (1.45 GHz–3.9 GHz) and reach a reflection coefficient over −20 dB. Simulation and measurements both revealed two prominent resonance frequencies at around 1.8 GHz and 3.3 GHz. The measurements and simulations (including the HFSS, CST, and COMSOL Multiphysics results) were well in agreement (Figure 5d). In the simulation, the ideal properties of the material were considered, making the simulated results slightly different from the measurements. Cable and connection losses, as well as VNA calibration issues, may all have an impact on the experimental measurements. The combination of these factors is expected to lead to a slight mismatch between the results, but this is tolerable, since imaging is independent of the resonance frequency of a signal.

Some other frequency domain performance parameters, such as gain (dBi), efficiency, as well as 2D and 3D radiation patterns, were also simulated and analyzed. Despite the simulators using two different numerical integration methods (CST: finite integration technique, HFSS: finite element method, COMSOL Multiphysics: finite element method), they produced almost similar outcomes, ensuring antenna effectiveness for our application. The simulated findings were then matched with the measured results, and the figure demonstrates that the simulated outputs are nearly identical to the measured outputs, even though certain inconsequential variations occur as a consequence of other undesirable losses during manufacture. As shown in Figure 6a, antenna gain was simulated as a function of frequency. The simulated gain is relatively similar. There is a peak gain of around 6 dBi and an average gain of around 4.3 dBi in the range of 1.3–3.7 GHz, and the experimental peak gain of the manufactured antenna was measured to be almost 6 dBi at 3.7 GHz, which is adequate for microwave imaging, since the distance between the radiating element (antenna) and the target object (hemorrhagic tissue) is quite low. The proposed antenna, therefore, offers a relatively lightweight structure and a high gain. Performance tests of the various antenna designs in Table 2 demonstrate that the proposed design has the widest bandwidth in comparison to the other tested designs, achieves a high gain, and may be employed for increased penetration in head imaging. Large electromagnetic penetration waves are necessary for accurate medical diagnostics, since they enable the antennas to locate and detect deep abnormalities inside the cranium. It should be emphasized that, because head tissues are lossy and greatly attenuate electromagnetic signals, the penetration of EM waves inside the head is poor [10]. Furthermore, it has been shown that electromagnetic penetration decreases with distance or higher frequencies [32]. Moreover, it has been demonstrated that SAR is reduced with distance [32]. As a result, the antenna’s lower frequency band (1–2.5 GHz) can be used to identify deep targets inside the brain, while its higher band can detect anomalies inside the tissues of the head at closer ranges to the antenna. As a result, it is preferable to retain a high wave penetration, while keeping the maximum SAR within the allowed level (0.1409 W/kg) in our study. As can be seen in Figure 6b, the radiation efficiency of the proposed antenna is more than 90% when calculated using both the finite integration technique (FIT) and finite element method (FEM) approaches, and this efficiency is shown to be constant over the antenna’s usable frequency range. Practically, the efficiency of a good antenna is anywhere between 50 and 60% [33], whereas the simulated efficiency of this well-designed antenna is over 90%, and experimental radiation efficiency was measured to be as high as 85%. When comparing experimental and simulated results for efficiency and gain, there are only small discrepancies. This small variation between the two results may be attributed mostly to production defects. Another reason why the findings obtained via simulation are not the same as those obtained through measurement is, since the simulation takes into consideration the ideal properties of the material, whenever the copper elements that are layered over the FR-4 substrate display a certain amount of resistive behavior, which causes losses and results in a difference between the simulated and measured findings.

In Figure 7a,b, co-polarization and cross-polarization patterns are presented at a lower resonance frequency of 1.8 GHz and a higher resonance frequency of 3.3 GHz for XY and YZ planes (phi = 0 and 90°, respectively). According to the 2D pattern both measured and simulated, co-polarization radiation is considerably greater than that of cross-polarization. Despite its numerous side lobes, the antenna exhibits omnidirectional behavior. Figure 7 shows that, throughout its operating frequency range, the proposed antenna displays both an omnidirectional radiation pattern at phi = 90 and bidirectional characteristics at phi = 0°. It is also noticeable that cross-polarization is lower than −20 dB. To verify the stability and shape consistency of the radiation pattern, the radiation pattern was measured experimentally at several frequency points over the functional range, and the results were compared with the simulated results. As can be seen in Figure 7, the measured and simulated results correlate quite well, and the antenna displays a consistent omnidirectional radiation pattern over the operational frequency range. The 3D radiation pattern at 2.4 GHz is shown in Figure 7c to verify that the intended antenna provides omnidirectional characteristics. The result clearly demonstrates that the antenna can produce excellent omnidirectional characteristics.

### 3.3. Time-Domain Analysis

The antenna was evaluated in three different ways to determine its input/output signal properties and signal distortion, including face-to-face and side-to-side (both on the X and Y axes) input/output combinations, as seen in Figure 8a–c. It is imperative to determine whether an antenna performs effectively under MWI by analyzing its time-domain characteristics, including transmit–receive signal properties. In all three approaches, we considered a separation distance of 250 mm between transmitting and receiving antennas. However, the external media caused a distortion between the pulses sent and received. The receiver would be expected to detect the pulses transmitted by the transmitter. Assessing signal distortion, therefore, requires time-domain observations. As seen in Figure 8a,b, the signals received in the face-to-face and side-to-side (X-axis) cases were similar to the signals transmitted, in contrast to the side-to-side (Y-axis) radiation case. As a result, the use of face-to-face or side-to-side (X-axis) configurations for MWI zones is strongly encouraged. The group delay time parameter can also be used to characterize the ability of the antenna to maintain a stable pulse phase during transmission. The group delay (GD) of three different antenna combinations is shown in Figure 8d. As can be seen in the figure, the group delay variations for these three combinations were limited to 2 ns in the functional range at a 250 mm distance, where a maximum of 3.8 ns allowed for group delay variations in MWI [34].

### 3.4. The Proposed Antenna for Head Imaging

The head model used in this study is identical to the Specific Anthropomorphic Mannequin (SAM) head phantom specified in the IEEE 1528 standards for head-mounted mobile phone SAR testing [35], as shown in Figure 9a,b. A perfectly matched layer (PML) spherical air domain was created around the antenna and head phantom to prevent undesired reflection, which absorbs all outgoing waves and serves as an anechoic chamber to simulate testing in an infinitely wide-open environment.

Over the operating frequency range, the vector Helmholtz equation was used to calculate the EM wave interactions:(5)$$\nabla \times \frac{1}{\mu_{r}}\nabla \times E-k_{o}^{2}\varepsilon_{r}E=0$$
where **E** is the electric field, $\mu_{r}$ denotes the relative permeability, $k_{0}$ denotes the free-space wave vector, and $\varepsilon_{r}$ denotes the permittivity. When a human body is subjected to a radio frequency (RF) electromagnetic field, the SAR is the amount of energy absorbed per unit mass. The SAR is an important consideration for microwave devices to ensure operational safety when exposed to the human body [32]. Tissue absorption power is described as W/kg, which measures the power absorbed per mass of tissue. The SAR is normally calculated by averaging over the entire body or a small sample size (typically 1 g or 10 g of tissue). The SAR for electromagnetic energy can be determined from [32]:(6)$$\mathrm{SAR}=\sigma \frac{\left|E^{2}\right|}{\rho }$$
where σ denotes the electric conductivity of the tissue, |E| is the electric field magnitude, and ρ is the tissue density. The head phantom system setup with the proposed antenna design is shown in Figure 9. The phantom consists of the head with and without stroke. The head has an average permittivity of 45.8 and an average conductivity of 0.76 S/m [36]. The appropriate permittivity and conductivity values for a 20 mm radius sphere were employed to represent the stroke [7,37]. The electrical properties of hemorrhagic and ischemic strokes are distinct from one another; the permittivity and conductivity for hemorrhagic stroke are 61.065 and 1.583 S/m, respectively, and 30 and 0.5 S/m for ischemic stroke, respectively [38]. In this simulation, a single antenna was placed 40 mm from the stroke’s center. An iterative method, the general minimal residual method (GMRES), with an automatic Newton method, was utilized for the 3D models, which was the standard setting for the COMSOL finite element solver employed. As shown in Figure 9b, the 3D head model was meshed using a combination of tetrahedral and prism elements. The absolute and relative tolerances in the stationary solver were set to 0.01; additionally, second-order elements were employed because they showed better convergence results in comparison with first-order elements, as reported in earlier studies [39,40]. The total number of elements of the head model, including PML and far-field domains, was 912,476, which corresponds to 5,848,762 degrees of freedom. All simulations were conducted on an HP OMEN 30L workstation, which has a CoreTM 10700 CPU, with an internal clock frequency of 3.8 GHz and 64 GB of memory.

The reflection coefficient (S11) in free space, with a normal head, and with a head that includes either hemorrhagic or ischemic stroke, is shown in Figure 10. The Figure clearly shows that, compared to the free-space case, the reflection coefficient and the bandwidth were reduced due to the lossy and heavy dielectric properties of normal and abnormal head tissues. Moreover, results show a difference between the normal head and between the two types of strokes over the operating frequency range, reaching more than 0.4 dB at 1.2 GHz. Though the difference is less pronounced, consistent with other published studies [22,23,24,25], this finding reveals the capability of microwave imaging in differentiating between the normal and abnormal heads, and we assume that this difference will be more obvious when using more antennas and multiple head layers.

Figure 11 shows the antenna’s simulated gain in both free-space and head phantom setups. It shows that the gain is greater in the presence of a head phantom than in free space. According to the results, head phantom simulations yield a higher gain (>6 dBi at 3.7 GHz). Furthermore, in both free-space and head phantom simulations, the antenna demonstrates a strong gain across the bandwidth proportional to the increase in the simulated frequency.

The SAR distribution simulated using COMSOL Multiphysics for the normal head, head with hemorrhagic stroke, and head with ischemic stroke over the entire operating frequency range is shown in Figure 12a. The region of interest (ROI) of the measured SAR values is located at the region of the stroke. This figure clearly illustrates that there was a significant difference between the two types of strokes, with the model with hemorrhagic stroke yielding the highest amount of SAR, followed by the normal head, and finally the head with ischemic stroke. This finding supports the ability of microwave imaging to be an effective approach for detecting brain stroke. The highest SAR value was associated with hemorrhagic stroke and was 0.06 W/kg (1 g) at 1.3 GHz, which is almost double the SAR value of the head with ischemic stroke (0.03 W/kg). These SAR values were measured at the region of the stoke inside the head model. Furthermore, in all three cases, the maximum SAR value of the whole head model was 0.1409 W/kg, which is less than the magnitude of the IEEE public radiation exposure limit of 1.6 W/kg [23,35]. It is also less than values published in the literature, namely, 0.608 W/kg in [19], below 0.8 W/kg at 20 mm distance in [41], and below 0.5 W/kg for wearable antenna systems for medical diagnostics reported in [42]. Many studies have demonstrated that the SAR decreases with depth [25,42], hence, the low SAR value of this proposed antenna design enables the increase in input power to achieve more field penetration inside the head if desired, while assuring that the SAR is within the safe limit.

Figure 12b, Figure 13 and Figure 14 depict the simulated electric field distribution for a normal head, a head with hemorrhagic stroke, and a head with ischemic stroke. As seen in Figure 12b, the distinction between the three cases was more prominent, especially with an ischemic stroke, in which the difference reached 2 V/m, which is larger than that reported in a previous study (0.63 V/m) [7]. In addition to the aforementioned, this finding supports the use of microwave imaging as a highly effective method for stroke identification. In contrast to the SAR figure, the head with an ischemic stroke exhibited the largest electric field values along the whole frequency operating range, nearly 1.25 times the value produced by a head with hemorrhagic stroke. The electric field within the head phantom was presented in Figure 13 for all three cases; head slices show that a head with ischemic stroke exhibited the maximum electric field at 1.2 GHz, whereas the normal head and head with hemorrhagic stroke had comparable values at 2.74 GHz. The difference between the three cases is clearly seen in Figure 14, where there is a pronounced variation in electric field distribution between the normal head and both stroke types at the three operating frequencies.

### 3.5. Comparative Analysis

Compared to earlier studies, the designed antenna exhibits some important advantages. The FR-4 substrate, a readily accessible and inexpensive material, was used in the printing of our proposed design. The basic layer configuration design simplifies manufacturing and minimizes manufacturing failures related to soldering and welding. Several existing antennas are compared in Table 3. In this analysis, antenna dimension, bandwidth, fractional bandwidth (FBW), gain, efficiency, and the SAR are considered. It is possible to consider the antenna as a candidate for microwave imaging, since it has improved performance over existing antennas subject to bandwidth, stable gain and efficiency over functional frequency ranges, compact size, as well as head phantom sensitivity.

## 4. Conclusions

This study aimed to develop a planar monopole antenna to detect hemorrhagic and ischemic strokes using microwaves. The design has several prominent features, including a gain of about 6.15 dBi, radiation efficiency of over 90%, and a fractional bandwidth of 94.34%. Additionally, the acquired bandwidth spans 1.3 to 3.7 GHz, which is optimal, since lower frequency ranges of electromagnetic radiation may permeate further within cells with minimum attenuation. Experimental verification of the design was carried out on prototype antennas on FR-4 substrates. Experimental and computed results for the impedance bandwidth (S_11_) show a reasonable degree of consistency. The antenna exhibits minimal signal distortion when also examined in the time domain. The behavior of the antenna was validated when installed near live tissues for the detection and characterization of hemorrhagic or ischemic stroke. The antenna’s viability is shown in simulations using a realistic head model, where the SAR is found to be within acceptable limits (0.1409 W/kg). In future work, we aim to utilize a set of antennas around the head. Additionally, instead of using one head layer by averaging the electric properties, the next work will combine all head tissue layers, as each layer has its own dielectric properties. These enable us to obtain a more accurate and more in-depth investigation of using microwave imaging as a viable tool for brain stroke diagnosis. Moreover, it provides a reliable platform to investigate various issues related to the effectiveness of microwave imaging, such as a time-domain study with the presence of various head tissues, the study of the received signals at different antennas, and eligible image reconstruction methods.

## Figures and Tables

**Figure 1 sensors-22-07235-f001:**
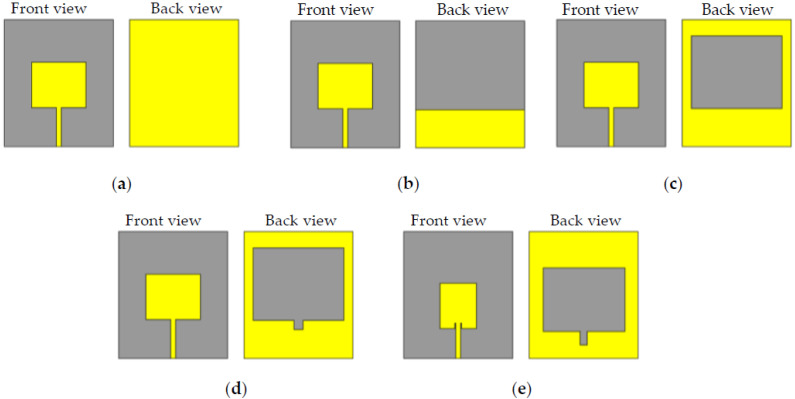
Several modifications of antenna patch and ground: (**a**) ordinary patch with full ground, (**b**) ordinary patch with partial ground, (**c**) ordinary patch with slotted ground, (**d**) ordinary patch with slotted ground and a notch parallel to the feed, (**e**) proposed final design structure.

**Figure 2 sensors-22-07235-f002:**
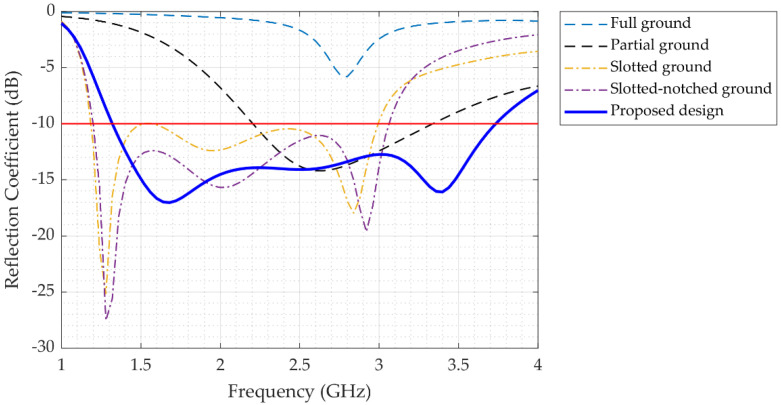
Reflection coefficient of the various proposed antenna design modifications shown in Figure 1.

**Figure 3 sensors-22-07235-f003:**
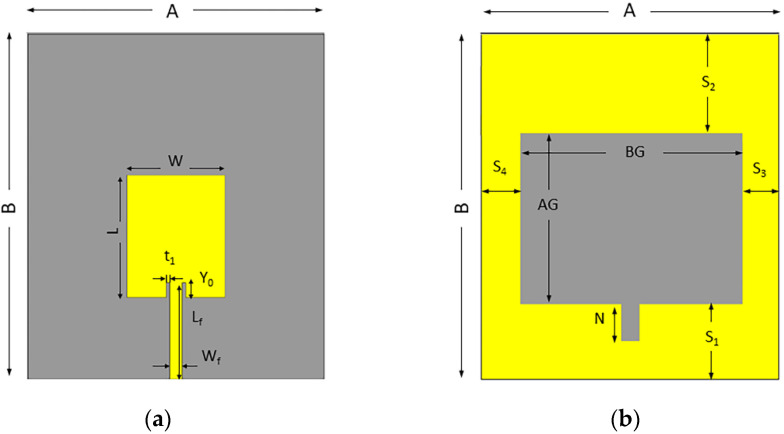
The geometric layout of the antenna: (**a**) front side, (**b**) back side. Parameter values for all dimensions are given in Table 1.

**Figure 4 sensors-22-07235-f004:**
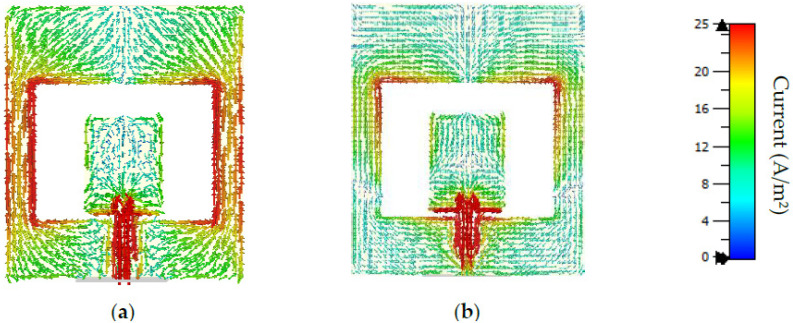
Flows of surface current of the proposed antenna in the functional range: (**a**) 1.8 GHz, (**b**) 3.3 GHz.

**Figure 5 sensors-22-07235-f005:**
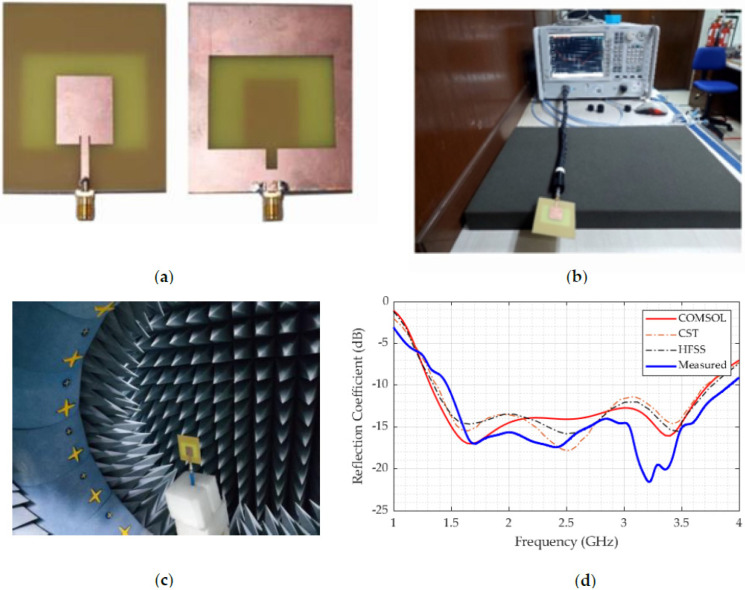
Fabrication and measurement: (**a**) Prototype front and back view, (**b**) −10 dB impedance bandwidth measurement, (**c**) Radiation pattern measurement setup in Satiomo star lab, and (**d**) Experimental and simulated impedance bandwidth (reflection coefficient).

**Figure 6 sensors-22-07235-f006:**
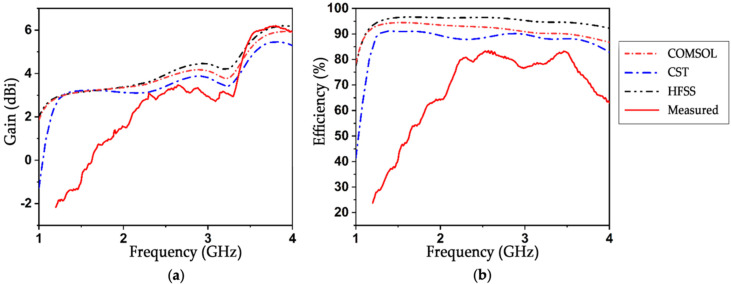
Performance analysis in terms of (**a**) Gain (dBi) and (**b**) Radiation efficiency (%).

**Figure 7 sensors-22-07235-f007:**
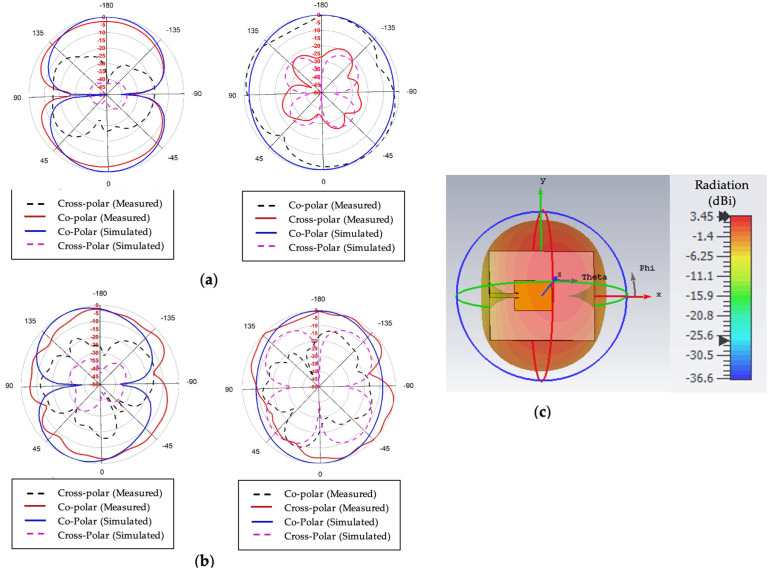
Measured and simulated 2D radiation pattern, including co-polarization and cross-polarization for phi = 0 and phi = 90: (**a**) at 1.8 GHz, (**b**) at 3.3 GHz, and (**c**) 3D radiation pattern.

**Figure 8 sensors-22-07235-f008:**
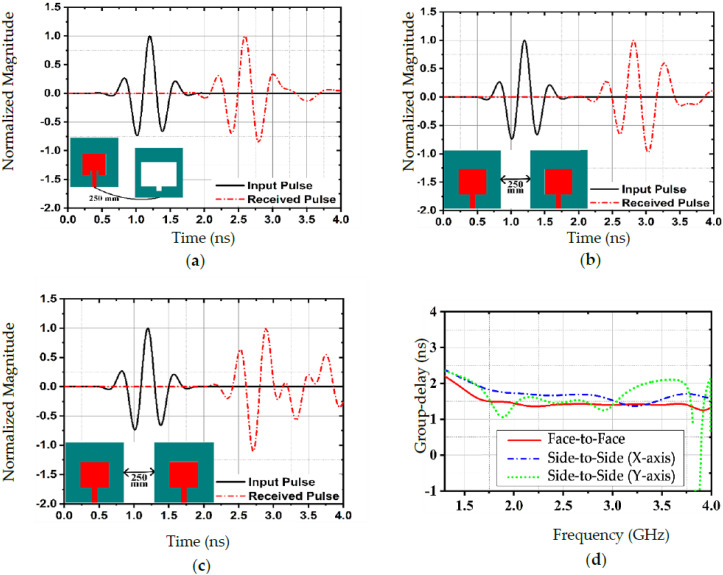
Normalized input/output signal magnitude for three different combinations: (**a**) Face-to-face, (**b**) Side-to-side [x-axis], and (**c**) Side-to-side [y-axis]. (**d**) shows the group delay of three different antenna combinations.

**Figure 9 sensors-22-07235-f009:**
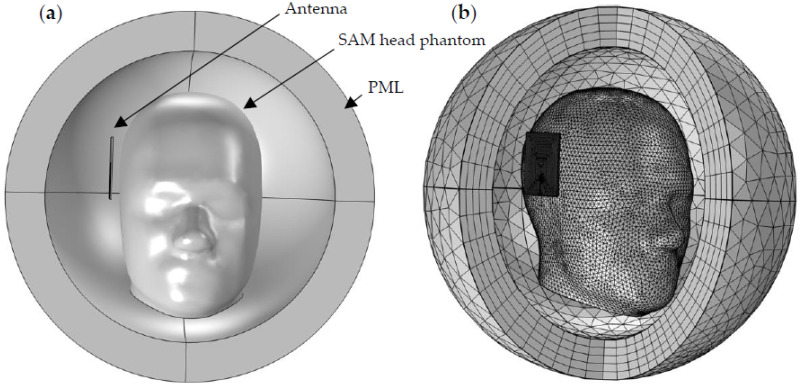
Model settings. (**a**) Front view of head model geometry encapsulated with a PML. (**b**) Model 3D mesh layout.

**Figure 10 sensors-22-07235-f010:**
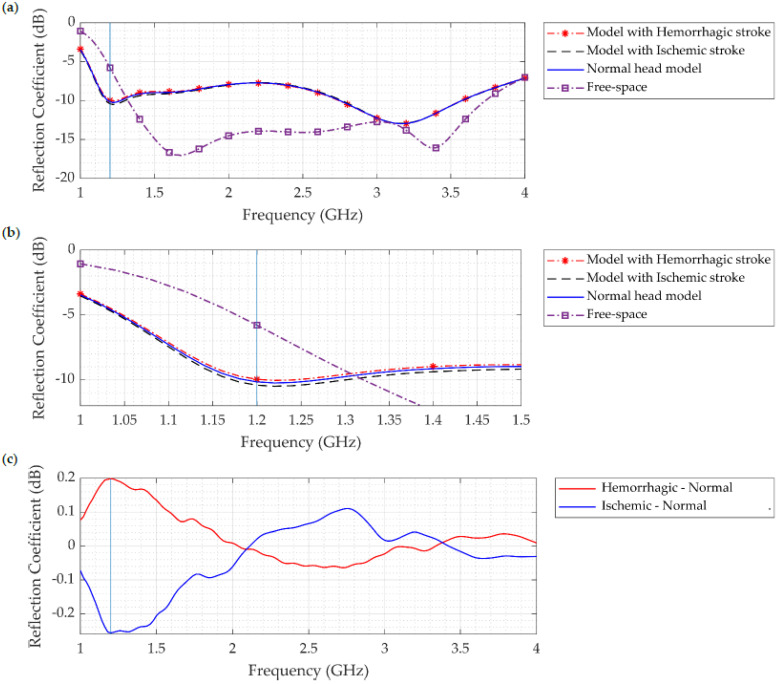
(**a**) Reflection coefficient (S11) of the normal head, head with ischemic stroke, head with hemorrhagic stroke, and free-space model. (**b**) Zoom of sub-figure (**a**) showing the differences in S11 at 1.2 GHz. (**c**) Difference in S11 between hemorrhagic stroke and normal head (red line), and ischemic stroke and normal head (blue line).

**Figure 11 sensors-22-07235-f011:**
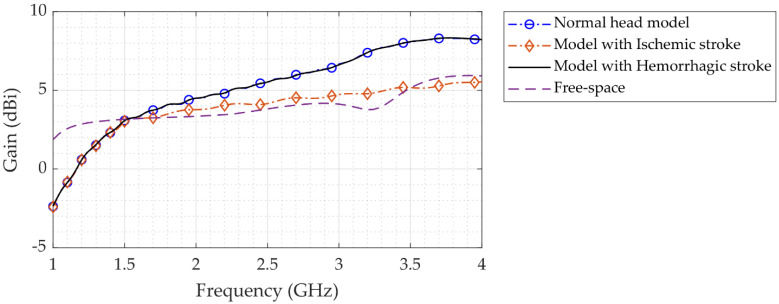
Simulated antenna gain in free space and with a head phantom.

**Figure 12 sensors-22-07235-f012:**
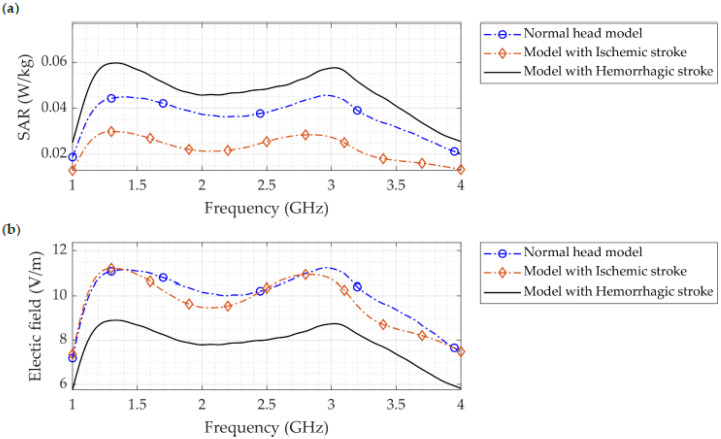
(**a**) SAR distribution where the stroke is located with the normal head, head with ischemic stroke, and head with hemorrhagic stroke inside the SAM head phantom. (**b**) Simulated electric field distribution for the normal model, model with hemorrhagic stroke, and model with ischemic stroke.

**Figure 13 sensors-22-07235-f013:**
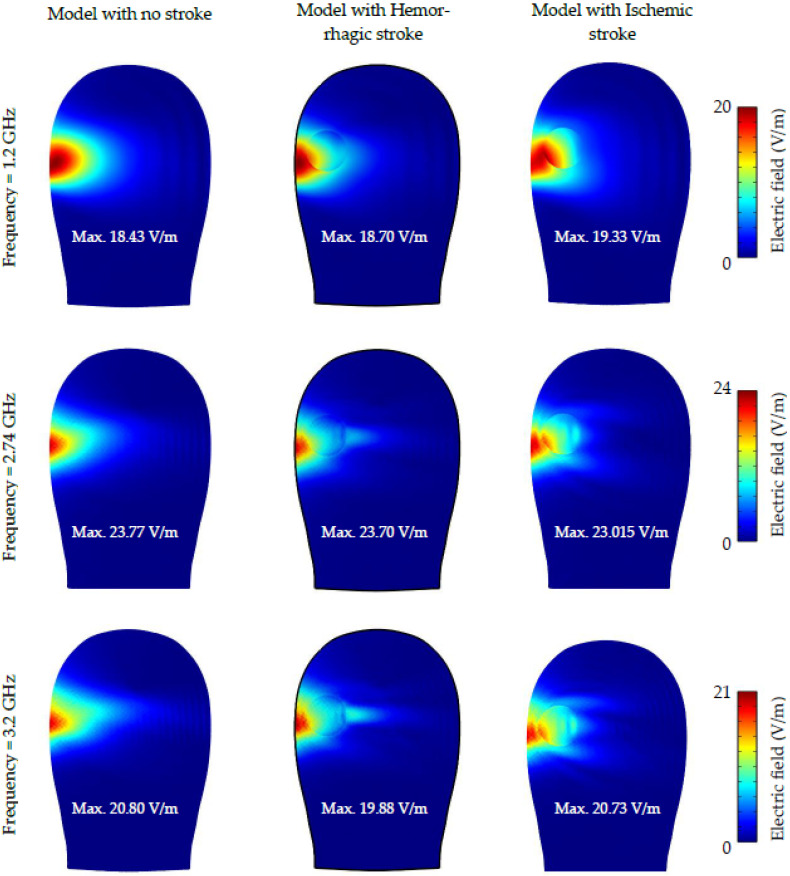
Slice plots of simulated electric field distribution for the case of the normal head, head with hemorrhagic stroke, and head with ischemic stroke at 1.2, 2.74, and 3.2 GHz.

**Figure 14 sensors-22-07235-f014:**
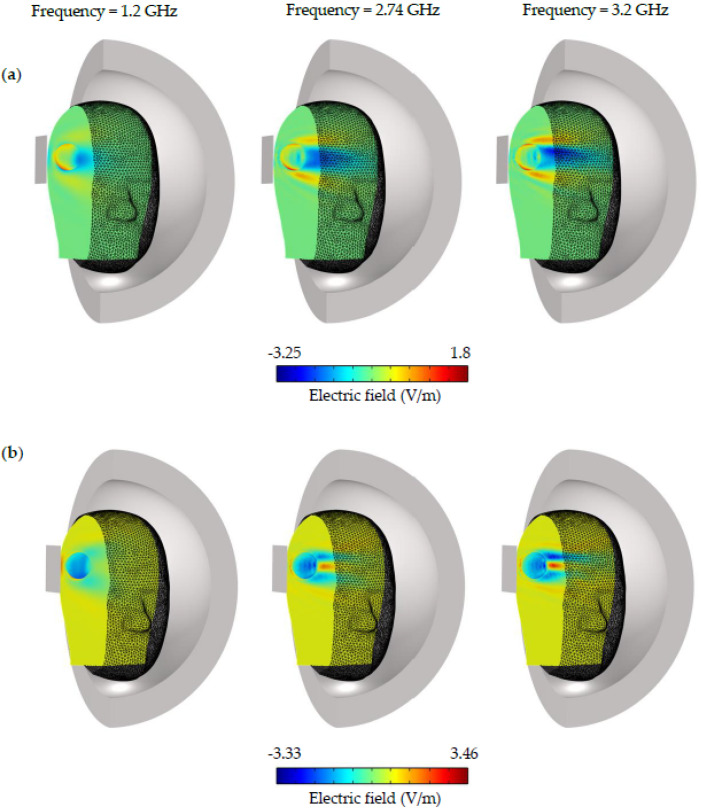
The electric field difference inside the SAM head phantom at 1.2, 2.74, and 3.2 GHz. (**a**) Electric field difference between the head with ischemic stroke and the normal model. (**b**) Electric field difference between the head with hemorrhagic stroke and the normal model.

**Table 1 sensors-22-07235-t001:** Values for all design-optimized parameters of the proposed antenna.

Parameter	Value (mm)	Parameter	Value (mm)
A	70	AG	35
B	60	BG	45
L	25	N	7.5
W	20	S_1_	15
L_f_	19.5	S_2_	20
W_f_	2.72	S_3_	7.2
Y_0_	3	S_4_	7.8
t_1_	0.5		

**Table 2 sensors-22-07235-t002:** Frequency range, bandwidth, and maximum gain for all the various tested antenna design modifications.

Modification	Operational Frequency (GHz)	Bandwidth (GHz)	Maximum Gain (dBi)
Full patch with full ground	No band	No band	N/A
Full patch with partial ground	2.2–3.3	1.1	3.3101
Full patch with slotted ground	1.2–3	1.8	5.4508
Full patch with full ground and notch	1.2–3.1	1.9	5.118
Proposed design	1.3–3.7	2.4	6.15

**Table 3 sensors-22-07235-t003:** Analyzing the proposed antenna against other novel designs.

Year	Reference	Dimension (λ^3^)	Bandwidth (GHz)	Fractional Bandwidth (%)	Gain (dBi)	Efficiency (%)	Maximum SAR (W/kg)
2019	[10]	0.083 × 0.09 × 0.018	1.00–2.00	66.6%	N/R	N/R	N/R
2016	[22]	0.30 × 0.08 × 0.02	1.10–2.20	66.67%	4.60	>70	N/R
2018	[25]	0.33 × 0.23 × 0.015	1.16–1.94	50%	N/R	83	0.500
2020	[29]	0.26 × 0.22 × 0.04	1.00–2.00	66.6%	Avg. 3.00	N/R	N/R
2021	[24]	0.277 × 0.238 × 0.006	1.19–3.56	100%	5.95	85	0.927
2020	[30]	0.253 × 0.19 × 0.010	1.9–3.6	61.81%	3.5	<90	N/R
2020	[28]	0.263 × 0.227 × 0.004	1.0–2.0	66.6%	N/R	N/R	0.45
2021	[27]	0.261 × 0.111 × 0.052	1.12–2.50	76.24%	>4	>70	>0.05
2020	[26]	0.283 × 0.250 × 0.009	1.70–3.71	74.4%	5.6	90	0.0023
2021	[31]	0.49 × 0.49 × 0.02	2.20–2.80	24%	4.7	<90	0.24
2022	**Proposed**	0.32 × 0.28 × 0.007	**1.3–3.7**	**94.34%**	**6.15**	**>90**	**0.1409**

## Data Availability

Not applicable.

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
