# Peer review of "Slotted Monopole Patch Antenna for Microwave-Based Head Imaging Applications"

_sensors, 2022, doi:10.3390/s22197235_

Round 1
Reviewer 1 Report
A Slotted Monopole Patch Antenna for Microwave-Based Head Imaging Applications has been proposed. Several modified antenna models based on the basic rectangular patch were designed and simulated. The single modified antenna unit has been fabricated. The proposed antenna exhibited reasonable performance and S11 measurement results shown agreement with the simulated results. The presented work has certain merit and contributing at some extent to the microwave antenna designing field. The work must need to be further improved based on the suggestions given as under.
(1) The overall English writing of the article should be improved. Pls carefully check the grammatical typos, articles usage and punctuation mistakes throughout the article before resubmitting it.
(2) In the first sentence of the abstract line 26, the authors wrote a sentence i.e. “An innovative monopole patch antenna for microwave-based hemorrhagic or ischemic stroke recognition is presented in this article.” The proposed antenna is very simple in design, I think the word “innovative” is not suitable to use here. I suggest authors to modify it as “new / modified”.
(3) In the abstract line 27, the sentence “The top and back radiators are developed and fabricated” not supports the data provided in the article. Further, I can only see one fabricated prototype in the manuscript. Please modify it carefully in the revised article.
(4) Time domain analysis in terms of group delay of the proposed antenna should be provided.
(5) Key contributions and novelty of the work should be explained in the bullets right after “Introduction” Section 1 of the article.
(6) In sub-section 2.1, the mathematical formulation must be added. How the radiation box, ground plane and other copper elements were designed? Pls provide the more in-depth formulation related to the transmission line model of the patch antennas.
(7) Figure 4, the authors have shown the current distribution at 1.8 GHz, 2.4 GHz and 3.5 GHz. However, in the return loss performance of the proposed antenna (Figure 2), I only can see the two resonances 1.8 GHz and 3.3 GHz. At what criteria the authors show three resonances? Pls give strong and solid justification regarding this and also explain it in the manuscript.
(8) Figure 6 (a) – (b), the authors has extracted the simulation results of gain and efficiency by using HFSS, COMSOL and CST. However, I didn’t see the measurement results of the manufactured antenna. It is strongly suggested to show the measured results of gain and efficiency of the proposed antenna.
(9) Pls provide the photographs of the fabricated antenna during the measurement, i.e. fixed inside anechoic chamber while measuring the radiation patterns.
(10) Provide the graphical representation of the overall measurement setup
(11) Figure 7 (a) – (b), again the measurement results of 2D far-field radiation pattern are not presented. Pls provide the measurement results of radiation patterns. Why the cross polar pattern performance at phi = 0 (around -20 dB) is lower than at phi = 90 (around -35 dB)?
(12) The future directions of the study should be explained the conclusion section of the article.

Author Response
We are pleased to resubmit for publication the revised version of Manuscript ID 1868896 entitled “Slotted Monopole Patch Antenna for Microwave-Based Head Imaging Applications”. We appreciated the constructive criticisms of the reviewers. We have addressed each of their concerns as outlined below.

Reviewer 2 Report
1- Line 105: "Referring to Figure1(a) ..." : You could use the inset fed microstrip antenna for better impedance matching. So
A- the statement you mentioned here is not valid. i.e. what you mentioned about why the S11 cannot be better than -10 is not valid.
B-This comparison is not fair for the last proposed antenna in Fig.1e compared to options in Fig.1a-d, Since you could use inset fed patch antenna and make it matched to 50 ohm.
2-Line 112: You may need to explain what is the reason for multiple resonance in Fig.1e. Also you may need to check how many modes are there in that bandwidth.
3-Line 145: What do you mean by "improved radiation pattern polarization"? You mean the polarization consistency or something else? Needs more details.
4-Line 146-147, From "The top radiator, ..." : It is expected because what is shown is vector plot is the current density and since the feed part is narrower than the patch then we see higher concentration. Tbh this sentence doesn't have any thing to say. We need more explanation in clearer way.
5- Line 155-157: To me the simulation with different toolboxes is abundant when comparing with real measurement. But respecting to the authors efforts it's up to them to keep it or just let it remain in the text as is.
6- Line 179-181, From "Which is ...": Did you mention somewhere if the near field or far field is being utilized in MWI? This is a vital information for the readers so they can differentiate near field MRI techniques and MWI?
If this is in the near field then Authors need to explain more about the significance of knowing about "antenna gain".
7- Lines 182 - 184, From "Performance test ...": "Increased penetration" mentioned here needs to be more discussed due the safety issues.
Also as a suggestion: what do you mean by "May" in "May be employed for increased penetration"? We dont have "May" if something is scientifically valid. It "will" or it "will not".
8- Line 186-187, From "Practically, ...": Do you have any reference for this claim about the acceptable efficiency of the antenna? Please mention that in the text.
9- Line 191-194: This radiation patterns needs to be presented for the lowest and highest frequency of operation too. To show pattern shape consistency? Just one frequency not acceptable when you are claiming UWB feature as an advantage.
10- Line 212-221: Authors need to explain more (or introduce more references) for the purposes of this time domain study. Also, Is this study done when a tissue (human gray/white matter or CSF) was around the antennas? or it was done in free space? what will the results look like if it was done next to human tissues.
11- Line 247: Is it realistic to assume the 4 cm diameter stroke area in the model? Is there any reference for that? What is the self resonance of the sphere in these three different cases: Normal, hemorrhagic and ischemic? These intrinsic resonances could help on understanding the response of the stroke area.
12- Line 256: 0.4 dB of difference can be due to many thing such as noise, simulation mesh calculations or etc. etc. and it might not be related to the different types of strokes. You need to have a tabular comparison of other methods with your own method. If they are not even close then this study might not be that significant. [Sorry that I am saying this].
Moreover, If you do this study in a heterogeneous model like human head, then you may get different results.
13- Line 272, Fig 11 : Is there any explanation, how come at the presence of head, the S11 is less matched, but the gain is higher? Is that because the pattern is more directed? The pattern study after and before the stroke should be presented too.
14- Line 283-295: What is the region of interest (ROI) in your SAR calculation compared other studies. And also please note if the SAR calculation you are presenting is normalized to the input E field or power. If it is not normalized then all your comparison you did with other works could be invalid.
Thanks.
Author Response

(The authors gave the same response as above.)

Round 2
Reviewer 1 Report
Thanks for your efforts to further modify the scientific work according to the suggestion. Moreover, there is still need to add literature and i strongly suggest you to cite the following quality papers related to your topic in the final revised manuscript.
https://doi.org/10.23919/ISAP47258.2021.9614427.
https://doi.org/10.1109/LAWP.2013.2237744.
https://doi.org/10.1109/APUSNCURSINRSM.2018.8608536.
https://doi.org/10.47037/2021.ACES.J.360717935
https://doi.org/10.1109/APUSNCURSINRSM.2017.8073120.
https://doi.org/10.1109/VTC2022-Spring54318.2022.9860542
https://doi.org/10.1109/iWEM.2018.8536682.
Overall quality of the manuscript looks fine now, however you need to further carefully proof read it and provide the good quality figures.
Author Response
We are pleased to resubmit for publication the revised version of Manuscript ID 1868896 entitled “Slotted Monopole Patch Antenna for Microwave-Based Head Imaging Applications”. We appreciated valuable comments from reviewer and academic editor. Kindly see the attached responses to these comments.
